# Full-range birefringence control with piezoelectric MEMS-based metasurfaces

Chao Meng [1,3], Paul C. V. Thrane [1,2,3], Fei Ding [1✉] & Sergey I. Bozhevolnyi [1✉]

Dynamic polarization control is crucial for emerging highly integrated photonic systems with diverse metasurfaces being explored for its realization, but efficient, fast, and broadband operation remains a cumbersome challenge. While efficient optical metasurfaces (OMSs) involving liquid crystals suffer from inherently slow responses, other OMS realizations are limited either in the operating wavelength range (due to resonances involved) or in the range of birefringence tuning. Capitalizing on our development of piezoelectric micro-electro-mechanical system (MEMS) based dynamic OMSs, we demonstrate reflective MEMS-OMS dynamic wave plates (DWPs) with high polarization conversion efficiencies (~75%), broadband operation (~100 nm near the operating wavelength of 800 nm), fast responses (<0.4 milliseconds) and full-range birefringence control that enables completely encircling the Poincaré sphere along trajectories determined by the incident light polarization and DWP orientation. Demonstrated complete electrical control over light polarization opens new avenues in further integration and miniaturization of optical networks and systems.

[1] Centre for Nano Optics, University of Southern Denmark, Campusvej 55, Odense DK-5230, Denmark. [2] SINTEF Microsystems and Nanotechnology, Gaustadalleen 23C, 0737 Oslo, Norway. [3] These authors contributed equally: Chao Meng, Paul C. V. Thrane. ✉email: feid@mci.sdu.dk; seib@mci.sdu.dk

Recent years brought explosive growth of OMS applications in terms of both diversity and sophistication[1], especially with OMS configurations becoming dynamic, i.e., offering not only spatial but also temporal control over transmitted and reflected optical fields[2]. Plentiful dynamic OMSs have been demonstrated in the past decade, implementing active amplitude and phase modulation with diversified functionalities[3–10]. Despite significant progress achieved in dynamic control of amplitude and phase responses, OMS-enabled active polarization control in the optical range remains largely unexplored. At the same time, the polarization of light is crucial for the realization of adaptive integrated photonic systems due to the fundamental role of polarization as another intrinsic characteristic of optical waves, which is uncorrelated with the amplitude and phase. To implement active polarization control with dynamic OMSs, one should be able to dynamically tune the OMS optical anisotropy in a wide range. The corresponding OMS realizations rely on the birefringence property of natural materials[11,12] or metasurfaces[13–18], consisting of anisotropic meta-atoms. In either case, since available OMS thickness (i.e., interaction length) and birefringence are limited, identification of a suitable OMS configuration is a rather complicated and arduous issue, with the outcome exhibiting drawbacks, such as slow responses, narrow bandwidths, limited dynamic ranges of birefringence, and low polarization conversion efficiencies (Supplementary Table 1).

In this article, we demonstrate electrically controlled full-range birefringence by integrating a plasmonic OMS with judiciously designed anisotropic meta-atoms and a thin-film piezoelectric MEMS mirror. The resulting MEMS-OMS-based DWP operates in reflection, featuring continuously varied anisotropy and enabling complete encircling of the Poincaré sphere, e.g., the polarization conversion from linear to circular, orthogonal linear, and opposite circular polarizations (Fig. 1). The OMS component, which is composed of a glass substrate with an OMS layer containing a 2D array of identical rectangular gold nanobricks, is placed close to an actuated MEMS gold mirror, whose separation $T_a$ from the OMS is accurately controlled by applying the actuation voltage $V_m$ (Fig. 1a). In the conceptual schematic, a linearly polarized light beam is incident onto the DWP, and the polarization state of the reflected light is shown continuously adjusted by changing the actuation voltage, with different polarization trajectories encircling the Poincaré sphere being available for different DWP orientations (Fig. 1a).

## Results

### MEMS-OMS-based DWP design

Variable birefringence of the proposed DWP originates from the anisotropy of OMS elements placed at different distances from the MEMS mirror and can be described by considering the hybrid OMS-integrated Fabry–Pérot (FP) configuration[19–21]. For example, large reflection amplitudes along with large phase differences (>270°) for two orthogonal linear polarizations (LPs) can be realized for the operation wavelength $\lambda = 800$ nm at a specific air gap of $T_a = 400$ nm for OMS nanobricks with sufficiently large aspect ratios (Fig. 1b). By varying the gap $T_a$, the phase difference, i.e., the DWP birefringence, can be dramatically changed, covering the whole tuning range of 0-2π, in a periodic fashion with the spatial period $\lambda/2 = 400$ nm (Fig. 1c). Note that the selected size of the OMS unit cell is $\Lambda = 250$ nm ($\Lambda < \lambda/2$), resulting in our choice of the nanobrick long side length $L_u = 200$ nm due to the electron-beam lithography (EBL) fabrication (aspect ratio) constraints. The short side length $L_v$ is determined by maximizing reflection amplitudes for both orthogonal polarizations (i.e., $|r_{uu}|$ and $|r_{vv}|$) expected at all air gaps $T_a$ along with ensuring the full-range birefringence control (Fig. 1c and Complementary Fig. S1). The nanobrick

height is also optimized to enable the full-range birefringence control and large reflection amplitudes, resulting in the choice of $t_m = 50$ nm, a value that conforms well with the fabrication tolerances (Supplementary Fig. S1). Variations in the reflected field amplitudes ($|r_{uu}|$, $|r_{vv}|$), phases ($\phi_{uu}$, $\phi_{vv}$), and birefringence ($\Delta\phi_{vu}$), when changing the air gap $T_a$ for the designed DWP unit cell ($L_u = 200$ nm, $L_v = 100$ nm), are shown in Fig. 1d, indicating the full 0-2π birefringence tunability and large reflection amplitudes (>0.9). Such a distinct polarization-dependent phase response of the designed DWP is achieved due to a very high OMS reflectivity for the light polarized along the long nanobrick side, so that the reflected field phase $\phi_{uu}$ does not depend on the air gap $T_a$, whereas the reflected field phase $\phi_{vv}$ of the orthogonal polarization varies periodically from 0 to 2π with changing the air gap (Supplementary Fig. S2).

### Performances of the DWP component

Following the simulations and design considerations presented above that indicate the possibility of realizing efficient DWPs with full-range birefringence tunability, we conducted the corresponding experimental investigations. The MEMS-OMS-based DWP for polarization conversion was assembled from a separately fabricated OMS (Fig. 1e) and an ultra-flat MEMS mirror that were wire bonded to a printed circuit board (PCB) for electrical connection (Supplementary Figs. S3, S4, Methods/Fabrication). Owing to the FP nature of the designed DWP component, the overall response is inherently periodic with respect to the variable air gap $T_a$, thus relaxing the requirement for small, nm-sized, air gaps essential for the previously developed MEMS-OMS configurations[22] and thereby simplifying the assembly process. We should however note that the access to nm-sized air gaps, although not easily gained, would allow one to operate the DWP with gap surface plasmons being generated[22], a regime that promises a slightly broader operation wavelength range as discussed below. To quantify the air gap $T_a$ at each actuation voltage $V_m$, we make use of the air gap-dependent MEMS-OMS operation[22] by exploiting the fact that the MEMS-OMS behaves as a mirror at periodically distributed air gaps (depending on the light wavelength used) regardless of the OMS design (Supplementary Figs. S5, S6). By determining the wavelength corresponding to the mirror-like operation (no polarization transformation) at each voltage, it is possible to accurately evaluate in situ the air gap $T_a$ realized in each polarization measurement (Supplementary Fig. S7).

For the characterization of the fabricated DWP performance, a supercontinuum laser beam with the well-defined LP state ($\theta_{LP} = 90°$) was directed onto the DWP oriented at $\theta_{DWP} = 45°$ (Fig. 2a). The MEMS mirror was electrically actuated to tune the DWP birefringence by changing the MEMS-OMS air gap and thereby adjusting the polarization state of the reflected light. Full Stokes polarimetry[23] was performed by rotating a quarter-wave plate (QWP) and a polarizer while analyzing the reflected light with a spectrometer (Methods/Characterization, Supplementary Fig. S8). Recording the DWP polarization-resolved reflection spectra (Supplementary Fig. S9) enables the determination of wavelength-resolved Stokes parameters along with the polarization conversion efficiencies and degree of polarization (DOP) while varying the MEMS-OMS air gap (Fig. 2b and Supplementary Figs. S9, S10). The quality of polarization conversion in the considered configuration and its operation bandwidth can be assessed by requiring efficient modulation of Stokes parameters $S_1$ and $S_3$, while keeping $S_2$ insignificant. It is seen (Fig. 2b, c) that the fabricated DWP performs very good, matching remarkably well the simulations, within the ~100 nm band (i.e., ~10%) around the design wavelength of $\lambda = 800$ nm. The experimental polarization conversion efficiencies (for all possible polarization conversions) of the DWP are rather high within the bandwidth

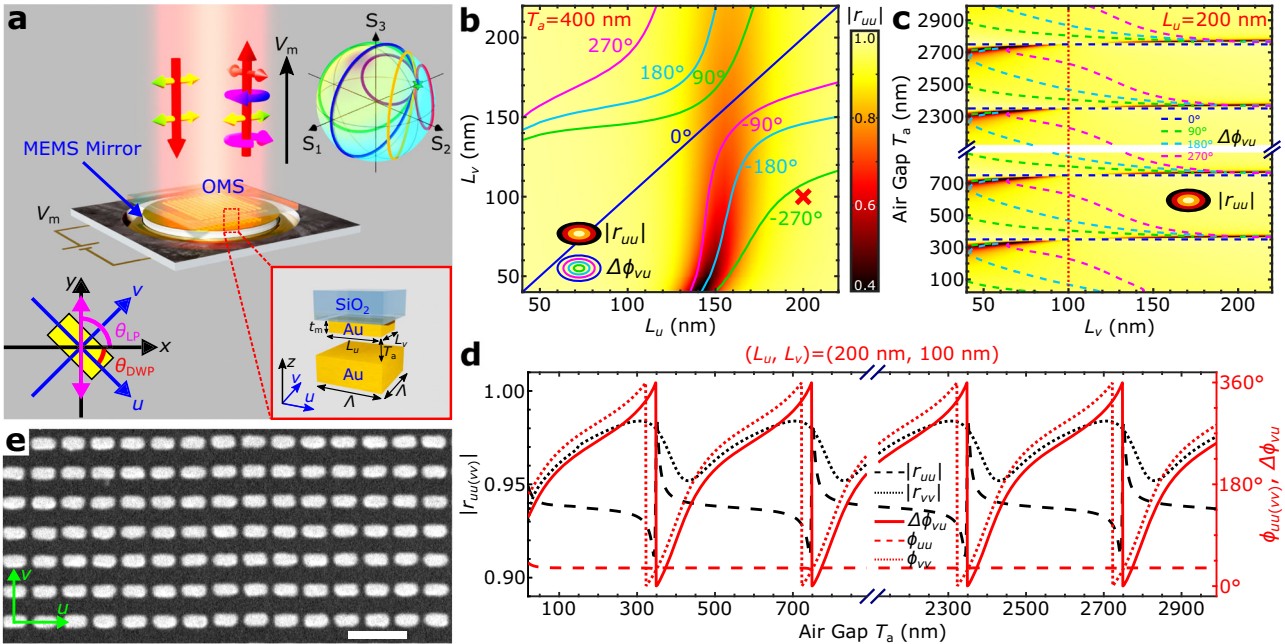

**Fig. 1 Electrically controlled MEMS-OMS-based DWP. a** Schematic of the dynamic wave plate (DWP) component consisting of an optical metasurface (OMS) on a glass substrate, mounted on a movable micro-electro-mechanical system (MEMS) mirror. By applying a voltage $V_m$, the separation between the MEMS mirror and OMS can be changed, thereby achieving a continuously variable birefringence in reflection. The bottom-right inset is a close-up view of the MEMS-OMS unit cell representing a gold nanobrick on glass separated by an air gap $T_a$ from the MEMS gold mirror. The top-right inset illustrates, with the Poincaré sphere, different reflected field polarization trajectories that can be realized by changing $V_m$ for the fixed incident linear polarization (LP) state $|y > (\theta_{LP} = 90°)$ and five different DWP orientations ($\theta_{DWP} = 15°, 30°, 45°, 60°, 75°$). The coordinate systems used are shown in the bottom-left inset. **b** Reflection amplitude ($|r_{uu}|$) and birefringence ($\Delta\phi_{vu}$) calculated as a function of the nanobrick side lengths $L_u$ and $L_v$, for the air gap $T_a = 400$ nm, with other parameters as follows: $\lambda = 800$ nm, $t_m = 50$ nm, $\Lambda = 250$ nm. Color map is related to the reflection amplitude, while colored lines represent constant birefringence contours. The red cross indicates the selected nanobrick dimensions $L_u \times L_v = 200$ nm × 100 nm. **c** Reflection amplitude ($|r_{uu}|$) and birefringence ($\Delta\phi_{vu}$) calculated as a function of the nanobrick side length $L_v$ and air gap $T_a$, for the other side length $L_u = 200$ nm. Color map is related to the reflection amplitude, while colored dashed lines represent constant birefringence contours. The vertical, red dotted line corresponds to the selected nanobrick dimensions $L_u \times L_v = 200$ nm × 100 nm. **d** Reflection amplitudes $|r_{uu(vv)}|$ (black dashed and dotted), phase $\phi_{uu(vv)}$ (red dashed and dotted), and birefringence $\Delta\phi_{vu}$ (red solid) calculated as a function of the air gap $T_a$ for the selected nanobrick dimensions $L_u \times L_v = 200$ nm × 100 nm. **e** Scanning electron microscopy (SEM) image of a part of the OMS nanobrick array fabricated on the glass substrate. The scale bar is 500 nm.

(Supplementary Fig. S9), being estimated at ~75% at the design wavelength $\lambda = 800$ nm (Fig. 2c), due to the full-range birefringence control realized and small absorption losses present in the MEMS-OMS configuration under off-resonance operation. Note that, as the air gap increases, the reflected light polarization changes periodically from incident LP $|y >$ state to $|r >$, $|x >$, $|l >$ states (Fig. 2a) with the corresponding Stokes parameters ($S_1$, $S_2$, $S_3$) changing from approximately $(-1, 0, 0)$ to $(0, 0, 1)$, $(1, 0, 0)$ and $(0, 0, -1)$, respectively (Fig. 2b, c). This DWP transformation is equivalent to the transformation from the mirror-like operation, or zero-wave plate (ZWP), to QWP, half-wave plate (HWP), and three-quarters-wave plate (TQWP). Note that the DOP remains being close to 1.0 during the whole DWP tuning, indicating an overall purity of polarization conversion. Slight deviations from the ideal conversion can be explained by fabricated nanobrick deviations from the design dimensions, the reflection from the front glass/air interface as well as possible mirror tilting. The polarization trajectory at $\lambda = 800$ nm follows one complete revolution around the Poincaré sphere, passing through basic polarization states $|y >$, $|r >$, $|x >$, and $|l >$, and circulating repeatedly with increasing $T_a$ (Fig. 2d and Supplementary Video S1). It is noticed that the polarization trajectory on the Poincaré sphere changing with the air gap $T_a$ exhibit a slight asymmetry (Fig. 2d and Supplementary Video S1), owing to different (also slowly varying) reflection amplitudes of the designed MEMS-OMS DWP structure for the two orthogonal linear polarizations (i.e., $|u >$ and $|v >$), as shown in Fig. 1d and Supplementary Fig. S2. The optimized DWP orientation, for generating all four typical polarization states

(i.e., $|x >$, $|y >$, $|r >$, $|l >$) in the best way, is $\theta_{DWP} = ~46°$ (Supplementary Fig. S11 and Table 2), since $|r_{uu}|$ is overall slightly smaller than $|r_{vv}|$ (Fig. 1d). The reflection from the front glass/air interface of the DWP component also matters, which, however, can be eliminated in future developments by making use of anti-reflection coating. As a final comment, in the current experimental demonstration, the DWP is operated with air gaps between 2 and 3 μm, covering two adjacent FP resonance orders and exhibiting practically the same performance.

High contrast between the main orthogonal polarization states ($|x >$ and $|y >$, $|r >$ and $|l >$), corresponding to HWP/ZWP and QWP/TQWP transformations, can be visualized at both the direct image (DI) and Fourier image (FI) planes (Fig. 3a, b). The dynamic polarization conversion behavior, induced by actuating the MEMS mirror with alternating voltages at a slow switching frequency, is captured by a charge-coupled device (CCD) camera (Supplementary Videos S2, S3), showing good repeatability and high modulation efficiency. By actuating the MEMS mirror with a periodic rectangular signal, filtering the reflected light with analyzers (i.e., $|x >$, $|y >$, $|r >$ or $|l >$) and detecting it with a fast photodetector, we observe relatively fast switching with rise/fall times of ~0.15/0.11 and ~0.36/0.23 ms for respective ZWP/HWP and QWP/TQWP transformations (Fig. 3c, d). The switching time for the QWP/TQWP transformation is approximately twice longer due to an approximately twice longer relocation of the MEMS mirror required (Fig. 2c), also reflected in an approximately twice larger switching voltage used (cf., Fig. 3c, d). Note

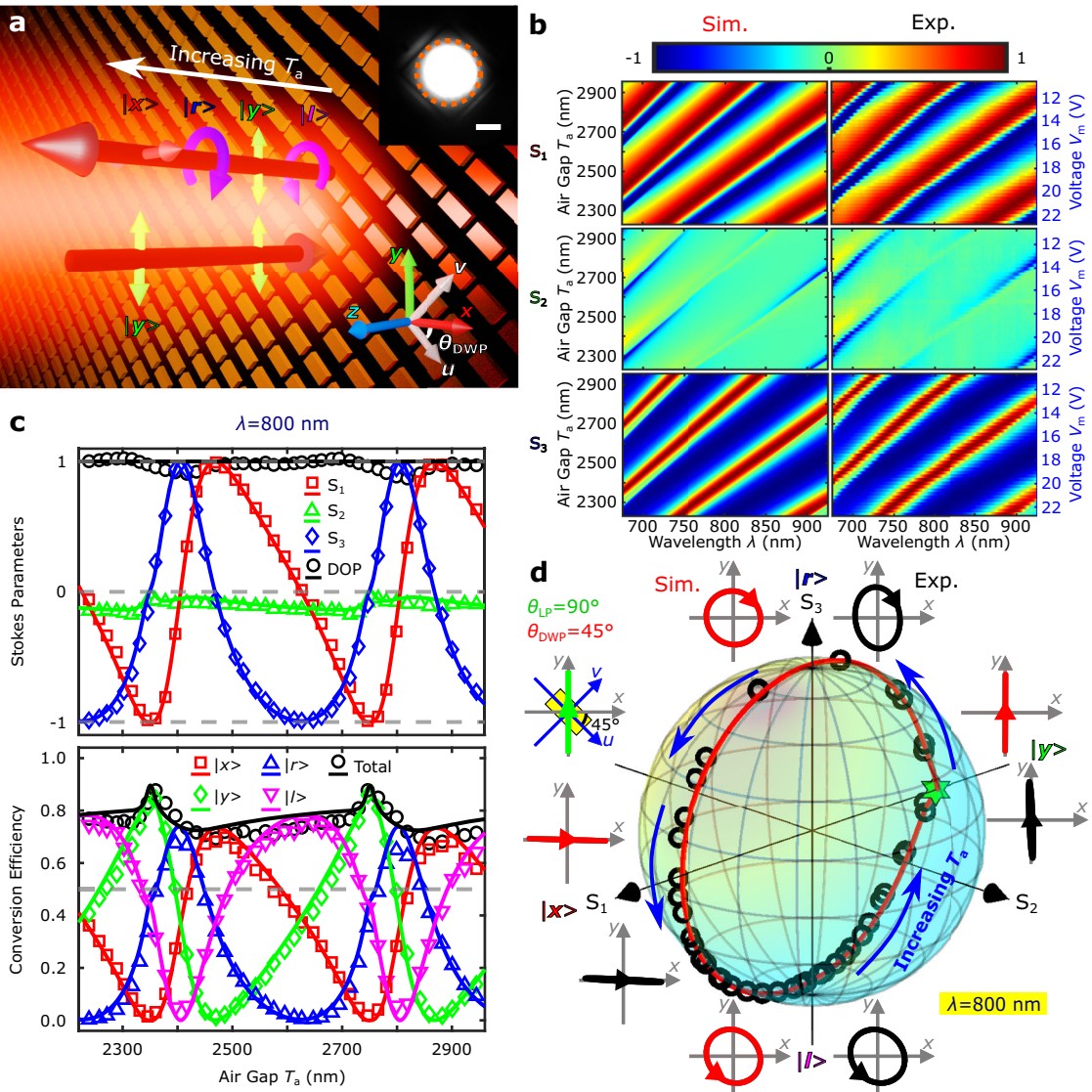

**Fig. 2 Polarization transformations for the incident LP oriented at 45° with respect to the DWP axes. a** Schematic of the dynamic wave plate (DWP) operation at the wavelength of 800 nm for the fixed incident linear polarization (LP) state $|y>$ ($\theta_{LP} = 90°$) and DWP orientation $\theta_{DWP} = 45°$, illustrating the reflected light polarizations being continuously changed from $|l>$, $|y>$, $|r>$ to $|x>$ with increasing the air gap $T_a$. The inset is a typical optical image of the optical metasurface (OMS) array illuminated with the incident laser spot confined within the orange dashed circle. The scale bar is 10 μm. **b** Calculated (left) and measured (right) normalized Stokes parameters ($S_1$, $S_2$, $S_3$) of the reflected light as a function of the air gap $T_a$ and light wavelength $\lambda$. **c** Calculated (lines) and measured (markers) normalized Stokes parameters (above) and polarization conversion efficiencies (below) as a function of the air gap $T_a$ for the light wavelength $\lambda = 800$ nm. **d** Calculated (line) and measured (circles) polarization trajectory on the Poincaré sphere mapping the reflected light polarization evolution shown in (**c**). As the air gap $T_a$ increases, the reflected polarization state revolves repeatedly around the sphere. The green star marks the incident LP state: $|y>$. The calculated (red) and measured (black) polarization ellipses illustrate the polarization contrast between the main orthogonal polarization states ($|x>$ and $|y>$, $|r>$ and $|l>$) realized by tuning the air gap $T_a$.

that, by optimizing MEMS mirrors for achieving faster switching, one should be able to reach switching frequencies in the MHz range[6,24]. We should also remark that a baseline voltage of ~14 V is used to move the MEMS mirror closer to the OMS, which would not be necessary for devices with more accurate mounting so that the maximum applied voltage can be kept below ~6 V.

**Versatile polarization transformations with DWP.** In addition to evaluating the DWP polarization tunability, polarization conversion efficiencies, and temporal response, we also explored the possibility of continuous LP rotation by a hybrid QWP-DWP component consisting of a general QWP, whose fast axis is oriented along *the x*-direction, and a DWP oriented at

$\theta_{DWP} = 45°$ (Fig. 4a and Supplementary Fig. S12). In this configuration, the QWP transforms the LP basis ($|u>$, $|v>$) of the DWP eigenstates into the CP basis (($|r>$, $|l>$) of the hybrid QWP-DWP component, thus enabling continuous LP rotation by tuning the DWP birefringence via tuning the air gap $T_a$. The reflected light polarization states measured while changing the air gap are visualized by a polarization trajectory around the equator of the Poincaré sphere (Fig. 4a, Supplementary Fig. S12, and Video S4), demonstrating the generation of LP light with arbitrary LP orientation by adjusting the actuation voltage $V_m$ (i.e., the air gap $T_a$). To ease steering potential practical applications, we have also investigated the polarization transformations realized for different DWP orientations ($\theta_{DWP} = 15°, 30°, 60°, 75°$) and fixed incident LP state $|y>$ (Fig. 4b, Supplementary Fig. S13,

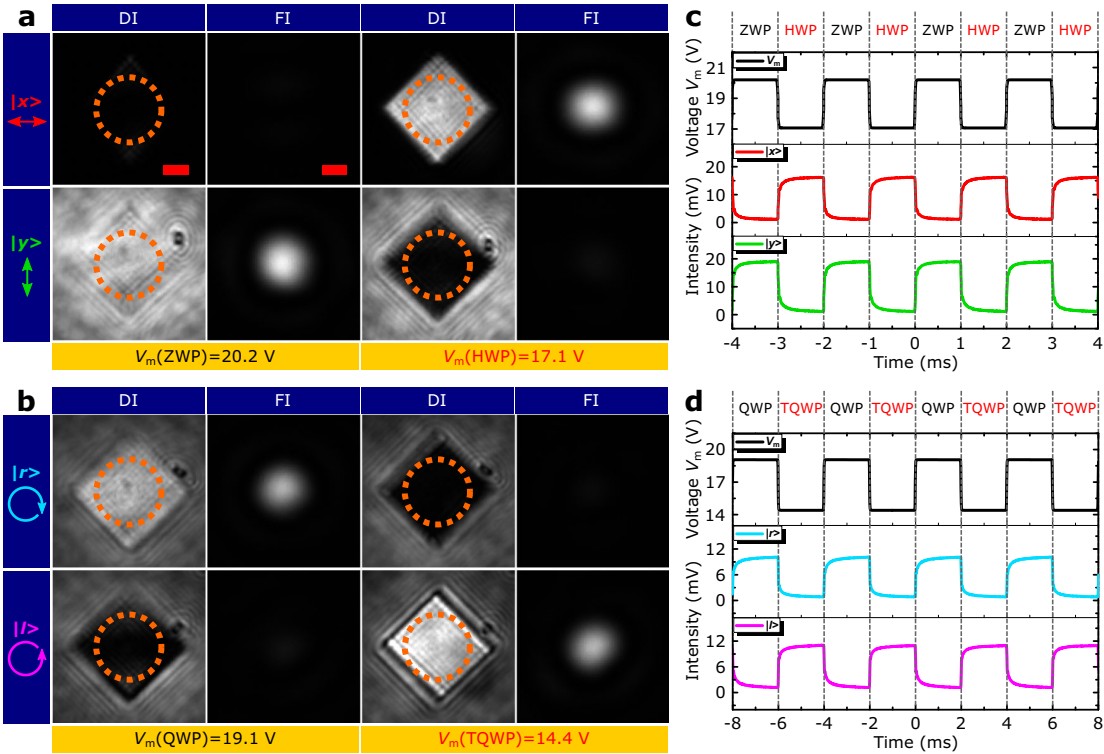

**Fig. 3 Polarization conversion dynamics. a, b** Optical images of the reflected light at the direct image (DI) and Fourier image (FI) planes for the fixed input linear polarization (LP) state $|y>$ ($\theta_{LP} = 90°$) and dynamic wave plate (DWP) orientation ($\theta_{DWP} = 45°$) at $\lambda = 800$ nm, showing polarization modulation between orthogonal (**a**) LP states ($|x>$, $|y>$), corresponding to zero-wave plate (ZWP)/half-wave plate (HWP) transformation, by changing the actuation voltage $V_m$ from 20.2 to 17.1 V, and (**b**) circular polarization (CP) states ($|r>$, $|l>$), corresponding to quarter-wave plate (QWP)/three-quarters-wave plate (TQWP) transformation, by changing the actuation voltage $V_m$ from 19.1 to 14.4 V. For the FIs, the light is filtered using an iris letting through only light reflected from inside the orange dashed circles (shown in DIs). The leftmost column in (**a, b**) indicates the polarizer orientations used for filtering the respective polarization states in the reflected light. Scale bars in the DIs and FIs are 10 μm and $0.02k_0$, respectively. **c, d** Temporal evolution of the reflected light power for **c** ZWP/HWP and **d** QWP/TQWP transformations, measured by actuating the MEMS mirror with a periodic rectangular signal and filtering respective polarization states.

and Video S5), as well as for different incident LP states ($\theta_{LP} = 60°$, 75°, 105°, 120°) and fixed DWP orientation $\theta_{DWP} = 45°$ (Fig. 4c, Supplementary Fig. S14, and Video S6). The polarization trajectories on the Poincaré sphere are seen to be differently tilted for different DWP orientations having the common point, the incident LP state $|y>$ (Fig. 4b), while being parallel to the plane ($S_1$, $S_3$) and reflecting the incident LP state (Fig. 4c). Note that, for any given point on the Poincaré sphere, one can identify suitable orientations of the DWP and incident LP state enabling closed polarization trajectories to pass this point, so that a multitude of polarization modulation capabilities can be realized with the same DWP. Finally, we would like to emphasize that all experimental results agree exceedingly well with the simulations without any fitting parameters, demonstrating convincingly that the fabricated DWP behaves according to the design and that the modeling approach developed is well suited for use in the future MEMS-OMS component developments.

## Discussion

To summarize, capitalizing on our development of the MEMS-OMS platform[22] we have demonstrated the electrically driven DWP operating in reflection with high polarization conversion efficiencies (~75%), broadband operation (~100 nm near the operating wavelength of 800 nm), fast responses (<0.4 ms) and full-range birefringence control that enables completely encircling the Poincaré sphere along trajectories determined by the incident light polarization and DWP orientation. It should be noted that, given the access to nm-sized air gaps, one can exploit the same design principle to realize the

DWP with gap surface plasmons being generated[22], a regime that promises a broader operation wavelength range (~160 nm) although probably at the expense of a lower efficiency (~50%)[22]. Importantly, the general approach developed can also be applied to design a DWP operating in transmission by using a partially transmitting MEMS mirror and placing an OMS in the middle of an FP cavity[25]. Given that a multitude of polarization modulation capabilities can be realized with the same DWP, we believe that the demonstrated electrically driven DWP configuration with full-range birefringence control opens fascinating perspectives for successful integration of high-performance compact dynamic polarization components into future miniaturized reconfigurable/adaptive optical networks and systems[26,27].

## Methods

**Numerical calculations**. All numerical simulations were done using COMSOL Multiphysics version 5.6. The model is composed of a rectangular volume with a square footprint with sides $\Lambda = 250$ nm and periodic boundary conditions were employed for both $u$ and $v$ directions. The DWP unit cell is divided into two parts of air and glass, with one gold nanobrick placed against the glass region. The corners of the nanobrick are rounded with a 5 nm radius. The refractive index of air is set as 1 and that of glass as 1.46 for all wavelengths, while the gold permittivity was interpolated as a function of wavelength from experimental tabulated values[28].

Using this model, the complex reflection and transmission coefficients for the glass/OMS/air interface are calculated for both propagation directions (i.e., a normal incident from glass or air) and for light linearly polarized along both $u$ and $v$ separately. These are used to calculate the total reflection coefficient $r_{FP}$ by including the gold substrate with the FP equation[19–21]

$$r_{FP} = r_{12} + \frac{t_{12}t_{21}r_{23}e^{i2\mathbf{k}_{n_2}T_a}}{1 - r_{21}r_{23}e^{i2\mathbf{k}_{n_2}T_a}} \qquad (1)$$

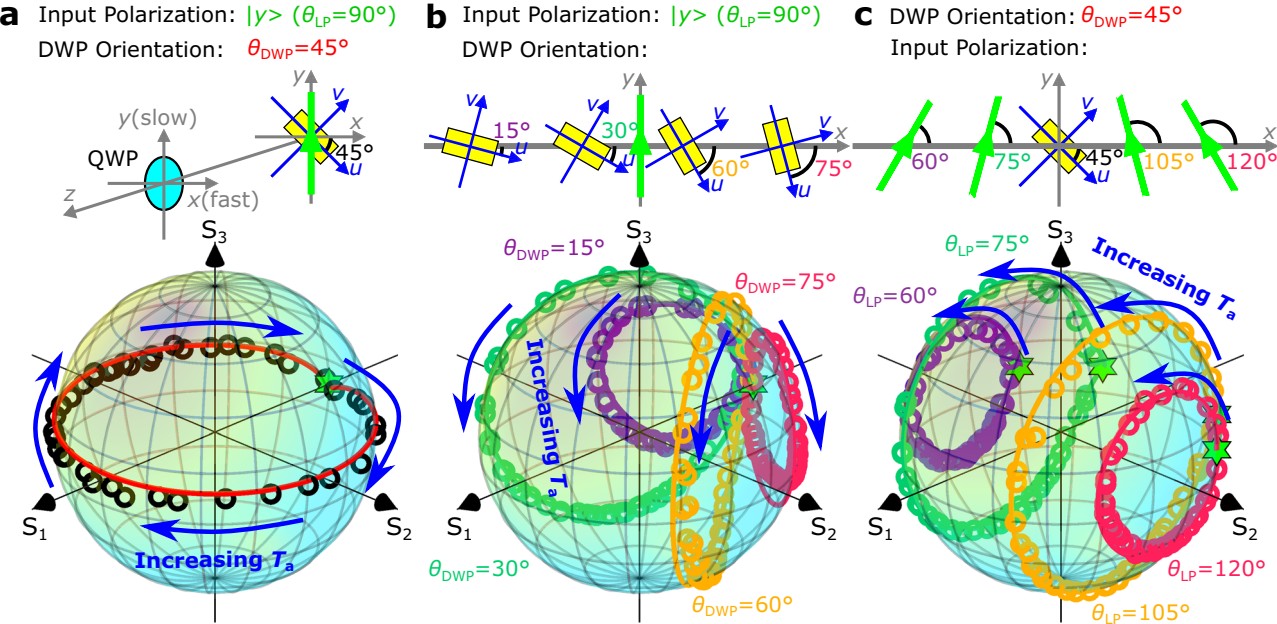

**Fig. 4 Versatile polarization transformations. a–c** Calculated (lines) and measured (circles) polarization trajectories on the Poincaré sphere realized at the wavelength of 800 nm by tuning the air gap $T_a$ for different dynamic wave plate (DWP) orientations, illustrating the diversity of possible polarization transformations: **a** continuous linear polarization (LP) rotation realized by combining the DWP with a conventional quarter-wave plate (QWP) with the incident LP state $|y>$ and DWP orientation $\theta_{DWP} = 45°$, **b** various elliptical polarization transformations realized for different DWP orientations with the fixed incident LP state $|y>$, and **c** various elliptical polarization transformations realized for different incident LP states with the fixed DWP orientation $\theta_{DWP} = 45°$. The green stars in (**a–c**) indicate respective incident LP states.

Here, $r_{mn}$ ($t_{mn}$) denotes the reflection (transmission) coefficients for light incident on material $n$ from material $m$ and the materials are numbered 1, 2, 3 for respectively glass substrate, air, and gold substrate, $n_2$ represents the refractive index of the air (i.e., medium 2 between the OMS layer and gold substrate), and **k** is the wavenumber in a vacuum. Note that the effect of the OMS on the interface between the air and glass substrate is included in $r_{12}$, $r_{21}$, $t_{12}$, and $t_{21}$, and that even for normal incidence $r_{FP}$ are polarization-dependent due to the anisotropy of the OMS layer. The reflection coefficient $r_{23}$ from the air/gold interface is calculated directly using the Fresnel equation. An illustration of the DWP geometry is shown in Supplementary Fig. S2 together with plots explaining the behavior of the total reflection coefficient $r_{FP}$ with varying $T_a$.

For air gaps much smaller than the wavelengths, there is near-field coupling between the nanobricks and the gold substrate in addition to the FP resonances[1,20], requiring numerical simulations including also the gold substrate in COMSOL. The results obtained using the FP equation were confirmed to give the same results as COMSOL simulations with the whole glass/nanobrick/air/gold substrate model when $T_a > 100$ nm for a wavelength of $\lambda = 800$ nm.

As a final comment, to compare with the measurements, the imaginary part of the gold permittivity is increased by three times in the simulations of Figs. 2, 4 and Supplementary Figs. S7, S9–S14, accounting for the surface roughness and grain boundary effects of the fabricated gold nanobricks as well as the increased damping associated with the titanium (Ti) adhesion layers between the gold/glass interface.

**Fabrication.** The OMS for developing MEMS-OMS DWP were fabricated using electron-beam lithography (EBL), thin-film deposition, and lift-off techniques. First, a 100-nm-thick poly(methyl methacrylate) (PMMA A2, MicroChem) layer and a 40-nm thick conductive polymer layer (AR-PC 5090, Allresist) were successively spin-coated on the glass substrate (Borofloat 33 wafer, Wafer Universe). Note that the glass substrate was preprocessed to have a 10-μm-high circular pedestal using optical lithography and wet etching, and the OMS pattern was defined on the pedestal by EBL (JEOL JSM-6500F field-emission SEM with a Raith Elphy Quantum lithography system). After development, the OMS layer was formulated by depositing a 1-nm Ti adhesion layer and a 50-nm gold layer (Tornado 400, Cryofox) followed by lift-off in acetone (Supplementary Fig. S4). The pedestal on the glass substrate is very effective for reducing the possible contaminants between the MEMS mirror and OMS surface, thus improving the stability and repeatability of the DWP components. The MEMS mirror is fabricated using standard semiconductor manufacturing processes (Supplementary Fig. S3), in which thin-film lead zirconate titanate (PZT) is incorporated for long-stroke, low-voltage electrical actuation. For use in the MEMS-OMS component, the ultra-flat MEMS mirror was sputtered with a 100 nm gold layer. After the gold deposition, the MEMS mirror surface is inspected with white light interferometry

(Zygo NewView 6300), showing overall good flatness and roughness all over the whole MEMS mirror (i.e., ~3 mm diameter) (Supplementary Fig. S3).

The MEM-OMS-based DWP component (Supplementary Fig. S4) was assembled by gluing the MEMS mirror with the glass substrate upon which OMS is structured and then glued to a printed circuit board (PCB), followed by a gold wire bonding process between the MEMS mirror and PCB for enabling simple electrical connection to a voltage controller used to actuate the MEMS mirror.

**Characterization.** The experimental setup is shown in Supplementary Fig. S8. A collimated fiber-coupled supercontinuum laser (SuperK Extreme, NKT) was directed through an HWP (AHWP10M-980, Thorlabs), a mirror, a linear polarizer (Pol₁; LPNIR050-MP2, Thorlabs), two beam splitters (BS₁,₂; CCM1-BS014, Thorlabs) successively, and then focused onto the DWP samples by an objective (Obj; M Plan Apo, ×20/0.42NA, Mitutoyo). The combination of HWP and Pol₁ is used for altering the input LP states as well as the intensity. The reflected light was collected by the same objective and passed through two beam splitters (BS₂,₃; CCM1-BS014, Thorlabs) and a tube lens (TL; TTL200-S8, Thorlabs), generating the first direct image plane where an iris is placed for filtering out the reflected light within the DWP area. The first direct image is then transformed by a relay lens (RL; AC254-200-B-ML, $f = 200$ mm, Thorlabs) to the corresponding Fourier image and captured by a CCD camera (CCD; DCC1545M, Thorlabs), according to a $2f$ configuration. Note that a flip lens (FL; AC254-100-B-ML, $f = 100$ mm, Thorlabs) is used for switching between the direct and Fourier images, and a Stokes analyzer composed of a QWP (AQWP10M-980, Thorlabs) and a linear polarizer (Pol₂; LPNIR050-MP2, Thorlabs) is implemented before the CCD camera for performing full Stokes polarimetry[23]. Two beam splitters are configured for cross-compensating the polarization-dependent phase shifts in the beam splitters for both incidence and reflection routes.

To obtain the wavelength-resolved full Stokes parameters, we replaced the CCD camera with a fiber-coupled spectrometer (QE Pro, Ocean Optics) and conducted measurements at the Fourier image plane. By rotating the QWP and Pol₂, we recorded polarization-resolved spectra of $I_x(\lambda)$, $I_y(\lambda)$, $I_a(\lambda)$, $I_b(\lambda)$, $I_r(\lambda)$, $I_l(\lambda)$, and the stokes parameters ($s_1$, $s_2$, $s_3$) are calculated as $s_1 = (I_x(\lambda) - I_y(\lambda))/(I_x(\lambda) + I_y(\lambda))$, $s_2 = (I_a(\lambda) - I_b(\lambda))/(I_x(\lambda) + I_y(\lambda))$, $s_3 = (I_r(\lambda) - I_l(\lambda))/(I_x(\lambda) + I_y(\lambda))$. For a reasonable comparison with the simulations, the stokes parameters ($s_1$, $s_2$, $s_3$) are normalized to the polarized proportion of the reflected light beam: $S_{1,2,3} = \frac{s_{1,2,3}}{DOP}$, and the degree of polarization (DOP) is defined as DOP $= \sqrt{s_1^2 + s_2^2 + s_3^2}$. The polarization conversion efficiency (Fig. 2 and Supplementary Figs. S9–S11) is defined as the ratio of the reflected light power in a specific polarization channel (i.e., $|x>$, $|y>$, $|a>$, $|b>$, $|r>$, $|l>$) to the incident linearly $|y>$ polarized light power.

The coordinates system used is indicated in the lower-left inset of Fig. 1a, with $z$ being the optical axis, $x$ and $y$ are transverse axes in the laboratory frame of

reference, while $u$ and $v$ are transverse axes oriented along the long and short sides of the rectangular nanobricks. The angle between $u$ and $x$ is denoted $\theta_{\mathrm{DWP}}$, while the angle between the $x$-axis and the polarization direction of LP incident light is denoted $\theta_{\mathrm{LP}}$.

To estimate the switching speeds between different orthogonal LP and CP bases (i.e., different DWP status), the setup described above is modified by replacing the input laser and CCD camera with a cw Ti:sapphire laser (Spectra-Physics 3900 S, wavelength range: 700 to 1000 nm), and a photodetector (PD; PDA20CS-EC, Thorlabs), respectively. The signals from the PD are acquired with an oscilloscope (DSOX2024A, Keysight). In the measurement, the MEMS-OMS-based DWP is actuated with periodically alternating voltages and different polarizations (i.e., $|x>$, $|y>$, $|r>$ and $|l>$) can be filtered by the Stokes analyzer.

## Data availability

All data that support the findings of the study are provided in the main text and Supplementary Information files. Raw data are available from the corresponding authors upon reasonable request.

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

## Acknowledgements

This research has received funding from the VKR Foundation (Award in Technical and Natural Sciences 2019, S.I.B. and Grant No. 37372, F.D.); the EU Horizon 2020 research and innovation program (Marie Skłodowska-Curie grant agreement No. 713694, C.M.); as well as from the Research Council of Norway (Project number 323322, P.C.V.T.). C.M. acknowledges Yao Xiao for the help in figure preparation, Ying Qu and Martin Thomaschewski for their help in the experiments. P.T. acknowledges Jon Vedum for helping with the control electronics for the MEMS mirrors.

## Author contributions

P.C.V.T. and C.M. performed the simulations and designed the OMS, fabricated and assembled the MEMS-OMS-based DWP samples. C.M. constructed the experimental setup, performed the measurements, and analysed the data. All authors contributed to the project idea, discussion of the results obtained and writing the manuscript. S.I.B. supervised the project.

## Competing interests

The paper authors along with Jo Gjessing and Christopher Dirdal from SINTEF are inventors on a related patent application filed by the University of Southern Denmark and SINTEF under United Kingdom Patent Application No. 2113182.6.
