## [Peer review file · Nature Communications]

REVIEWER COMMENTS

Reviewer #1 (Remarks to the Author):

This manuscript reports a design and validation of reflective MEMS-OMS waveplates that shows capabilities of dynamic polarization control. The authors provided a unique and reasonable design using the MEMS mirror and plasmonic metasurfaces, and achieved a high polarization conversion efficiency (about 75%), sub-millisecond response, and broad wavelength band of 100 nm at the center wavelength 800 nm. This manuscript reports numerical calculations and experimental results. The work is original and the results support their design and conclusions. This development is innovative in the nanophotonics and also has a broader impact as the design can be used in optical networks and other applications. The paper is well organized. Therefore, I support its publication in the Nature Communications.

Minor: Line 184, equation 1. It should define the n_2 in the equation.

Reviewer #2 (Remarks to the Author):

The paper titled "Full-range birefringence control with piezoelectric MEMS-based metasurfaces" shows the demonstration of a dynamic waveplate working in reflection based on the combination of an anisotropic metasurface with a MEMS mirror. Tuning the distance between the mirror and the metasurface the authors are able to control the birefringence along the two axes of the meta-atoms. They show that while the phase of light polarised along one side of the meta-atom is unbothered by the distance from the MEMS mirror, light polarised orthogonally to it acquires a phase which varies in a full $0-2\pi$ range, in a periodic fashion dictated by the cavity condition between the mirror and the metasurface. By now everybody knows the limitations of mechanical elements like MEMS in terms of speed and life cycles, but the authors never claim anything other than achieving a good degree of polarisation with a moderate modulation speed, which I think is truthful to what their experiment shows and could result in a generally useful tool for the community. I enjoyed reading the paper, it is very well written, and the concepts are explained clearly and with a good use of visuals. I enjoyed the videos particularly and wish they could somehow be part of the main text rather than the supplementary. I surely believe this paper deserves publication in Nature Communications. I only have one small comment that maybe the authors could address in their main text or detailed supplementary material. Both in figure 2 and in the supplementary video showing the polarisation trajectory along the Poincarè sphere changing with the air gap there is a clear asymmetry which is not acknowledged in the main text. This is even more evident in subsequent figures and videos and I'm assuming it is due to the difference in conversion efficiency that the device shows between the two orthogonal linear polarisations, plus

reflections from the glass surface. The authors do mention that an appropriate choice of angle of the metasurface + incident polarisation allows the waveplate to achieve any arbitrary state of polarisation. Since most experimentalists are going to be interested in using this device for 4 polarisation states, H, V, R and L, I would like the authors to provide the parameters needed to achieve these 4 states at the best of their capability with this device at its working wavelength.

As a final remark, I'm a bit doubtful about the comparison made in the supplementary table 1. The source of my doubt is that I am unsure how the authors have derived the efficiency in each of these experiments and if the comparison is fair. To evaluate the efficiency obtained in this manuscript the authors are using the degree of polarisation. Doing a quick search through the other cited works it seems to me that the numbers they provide in the table do not correspond to the degree of polarisation for all the other references and I think this might lead to a slightly unfair comparison. So I'd appreciate if the authors could clarify what they define as efficiency and use the same metric for the comparison of each experiment.

REVIEWER COMMENTS

Reviewer #1 (Remarks to the Author):

This manuscript reports a design and validation of reflective MEMS-OMS waveplates that shows capabilities of dynamic polarization control. The authors provided a unique and reasonable design using the MEMS mirror and plasmonic metasurfaces, and achieved a high polarization conversion efficiency (about 75%), sub-millisecond response, and broad wavelength band of 100 nm at the center wavelength 800 nm. This manuscript reports numerical calculations and experimental results. The work is original and the results support their design and conclusions. This development is innovative in the nanophotonics and also has a broader impact as the design can be used in optical networks and other applications. The paper is well organized. Therefore, I support its publication in the Nature Communications.

Minor: Line 184, equation 1. It should define the n_2 in the equation.

Reviewer #2 (Remarks to the Author):

The paper titled “Full-range birefringence control with piezoelectric MEMS-based metasurfaces” shows the demonstration of a dynamic waveplate working in reflection based on the combination of an anisotropic metasurface with a MEMS mirror. Tuning the distance between the mirror and the metasurface the authors are able to control the birefringence along the two axes of the meta-atoms. They show that while the phase of light polarised along one side of the meta-atom is unbothered by the distance from the MEMS mirror, light polarised orthogonally to it acquires a phase which varies in a full $0-2\pi$ range, in a periodic fashion dictated by the cavity condition between the mirror and the metasurface. By now everybody knows the limitations of mechanical elements like MEMS in terms of speed and life cycles, but the authors never claim anything other than achieving a good degree of polarisation with a moderate modulation speed, which I think is truthful to what their experiment shows and could result in a generally useful tool for the community. I enjoyed reading the paper, it is very well written, and the concepts are explained clearly and with a good use of visuals. I enjoyed the videos particularly and wish they could somehow be part of the main text rather than the supplementary. I surely believe this paper deserves publication in Nature Communications. I only have one small comment that maybe the authors could address in their main text or detailed supplementary material. Both in figure 2 and in the supplementary video showing the polarisation trajectory along the Poincaré sphere changing with the air gap there is a clear asymmetry which is not acknowledged in the main text. This is even more evident in subsequent figures and videos and I’m assuming it is due to the difference in conversion efficiency that the device shows between the two orthogonal linear polarisations, plus reflections from the glass surface. The authors do mention that an appropriate choice of angle of the metasurface + incident polarisation allows the waveplate to achieve any arbitrary state of polarisation. Since most experimentalists are going to be interested in using this device for 4 polarisation states, H, V, R and L, I would like the authors to provide the parameters needed to achieve these 4 states at the best of their capability with this device at its working wavelength. As a final remark, I’m a bit doubtful about the comparison made in the supplementary table 1. The source of my doubt is that I am unsure how the authors have derived the efficiency in each of these experiments and if the comparison is fair. To evaluate the efficiency obtained in this manuscript the authors are using the degree of polarisation. Doing a quick search through the other cited works it seems to me that the numbers they provide in the table do not correspond to the degree of polarisation for all the other references and I think this might lead to a slightly unfair comparison. So I’d appreciate if the authors could clarify what they define as efficiency and use the same metric for the comparison of each experiment.

Response to Reviewers

(listing all changes made)

We would like to thank the Nature Communications Editorial Team for providing us the opportunity to submit revised manuscript and supplementary materials, and both reviewers for their overall positive assessment of our work and very useful comments that help us further improve the manuscript. We have carefully considered all their comments and introduced changes to the main text and supplementary materials in response to most of the points raised.

We consider the revised manuscript to be improved and hope that it can now be accepted for the publication in *Nature Communications*.

Please, find below the point-by-point response to the reviewers' comments (original comments are highlighted in blue) and all changes made (apart from minor format adjustments).

Reviewer #1 (Remarks to the Author):

This manuscript reports a design and validation of reflective MEMS-OMS waveplates that shows capabilities of dynamic polarization control. The authors provided a unique and reasonable design using the MEMS mirror and plasmonic metasurfaces, and achieved a high polarization conversion efficiency (about 75%), sub-millisecond response, and broad wavelength band of 100 nm at the center wavelength 800 nm. This manuscript reports numerical calculations and experimental results. The work is original and the results support their design and conclusions. This development is innovative in the nanophotonics and also has a broader impact as the design can be used in optical networks and other applications. The paper is well organized. Therefore, I support its publication in the Nature Communications.

1. Minor: Line 184, equation 1. It should define the n_2 in the equation.

Our Response:

We thank the reviewer for noticing this inadvertent omission, and have added the definition of n_2 in the equation (1) accordingly (p. 9, revised manuscript):

“the materials are numbered 1, 2, 3 for respectively glass substrate, air and gold substrate, n_2 represents the refractive index of the air (i.e., medium 2 between the OMS layer and gold substrate), and k is the wavenumber in vacuum”.

Reviewer #2 (Remarks to the Author):

The paper titled “Full-range birefringence control with piezoelectric MEMS-based metasurfaces” shows the demonstration of a dynamic waveplate working in reflection based on the combination of an anisotropic metasurface with a MEMS mirror. Tuning the distance between the mirror and the metasurface the authors are able to control the birefringence along the two axes of the meta-atoms. They show that while the phase of light polarised along one side of the meta-atom is unbothered by the distance from the MEMS mirror, light polarised orthogonally to it acquires a phase which varies in a full $0-2\pi$ range, in a periodic fashion dictated by the cavity condition between the mirror and the metasurface. By now everybody knows the limitations of mechanical elements like MEMS in terms of speed and life cycles, but the authors never claim anything other than achieving a good degree of polarisation with a moderate modulation speed, which I think is truthful to what their experiment shows and could result in a generally useful tool for the community. I enjoyed reading the paper, it is very well written, and the concepts are explained clearly and with a good use of visuals. I enjoyed the videos particularly and wish they could somehow be part of the main text rather than the supplementary. I surely believe this paper deserves publication in Nature Communications.

1. I only have one small comment that maybe the authors could address in their main text or detailed supplementary material. Both in figure 2 and in the supplementary video showing the polarisation trajectory along the Poincarè sphere changing with the air gap there is a clear asymmetry which is not acknowledged in the main text. This is even more evident in subsequent figures and videos and I'm assuming it is due to the

difference in conversion efficiency that the device shows between the two orthogonal linear polarisations, plus reflections from the glass surface.

Our Response:

We thank the reviewer for bringing up this issue, which indeed requires additional clarification. Indeed, the asymmetry of the polarization trajectory along the Poincaré sphere is due to the difference in the reflection amplitude of the MEMS-OMS DWP structure for the two orthogonal linear polarizations. It should be noted that, in addition, this difference slowly changes with varying air gaps, as shown in Fig. 1d and Supplementary Fig. S2d. The reflection from the front air/glass interface partially matters here as well, which, however, can be eliminated in further developments by making use of anti-reflection coating.

Responding to this comment, we added relevant explanations (p. 6, revised manuscript):

“and circulating repeatedly with increasing T_a (Fig. 2d, Supplementary Video S1). It is noticed that the polarization trajectory on the Poincaré sphere changing with the air gap T_a exhibit a slight asymmetry (Fig. 2d, Supplementary Video S1), owing to different (also slowly varying) reflection amplitudes of the designed MEMS-OMS DWP structure for the two orthogonal linear polarizations (i.e., $|u\rangle$ and $|v\rangle$), as shown in Fig. 1d and Supplementary Fig. S2. The optimized DWP orientation, for generating all four typical polarization states (i.e., $|x\rangle$, $|y\rangle$, $|r\rangle$, $|l\rangle$) in the best way, is $\theta_{\text{DWP}} = \sim 46^\circ$ (Supplementary Fig. S11 and Table 2), since $|r_{uu}|$ is overall slightly smaller than $|r_{vv}|$ (Fig. 1d). The reflection from the front glass/air interface of the DWP component also matters, which, however, can be eliminated in future developments by making use of anti-reflection coating. As a final comment, in the current experimental demonstrations.....”.

2. The authors do mention that an appropriate choice of angle of the metasurface + incident polarisation allows the waveplate to achieve any arbitrary state of polarisation. Since most experimentalists are going to be interested in using this device for 4 polarisation states, H, V, R and L, I would like the authors to provide the parameters needed to achieve these 4 states at the best of their capability with this device at its working wavelength.

Our Response:

We thank the reviewers for encouraging us to check the possibility of achieving the typical 4 polarization states at the best of their capability. Owing to the non-equal reflection amplitudes of the MEMS-OMS DWP structure for the two orthogonal polarizations (as discussed in the answer to Comment 1), it is possible to get higher-purity of these 4 polarization states by optimizing the DWP orientation, when the incident light is fixed to $|y\rangle$ polarized. The optimized DWP orientation is $\theta_{\text{DWP}}=46^\circ$, which could be expected since $|r_{uu}|$ is slightly smaller than $|r_{vv}|$ (as shown in Fig. 1d). With DWP configured at $\theta_{\text{DWP}}=46^\circ$ and $|y\rangle$ polarized light incidence, higher purity $|x\rangle$, $|y\rangle$, $|r\rangle$ and $|l\rangle$ polarization states are evident with large modulation between S_1 and S_3 (almost from -1 to $+1$), while S_2 is negligible, as shown in Fig. R1 and Table R1. Regarding this comment, we added relevant sentences in the main text (see the answer to Comment 1), and Fig. S11, Table 2 into the Supplementary Materials (p. 12, revised supplementary information):

Figure R1. Polarization transformations for linearly $|y\rangle$ polarized light incidence and DWP oriented at respective $\theta_{\text{DWP}}=45^\circ$ and 46° . **a** Calculated normalized Stokes parameters (above) and polarization conversion efficiencies (below) as a function of the air gap T_a for the light wavelength $\lambda = 800$ nm, with $\theta_{\text{DWP}}=45^\circ$ and 46° , respectively. **b** Top view of the calculated polarization trajectory on the Poincaré sphere mapping the reflected light polarization evolution shown in **(a)**. The green star marks the incident LP state: $|y\rangle$. The solid and dashed lines are corresponding to DWP oriented at $\theta_{\text{DWP}}=45^\circ$ and 46° , respectively.

θ_{DWP}	45°				46°			
	S_1	S_2	S_3	Efficiency	S_1	S_2	S_3	Efficiency
$ x\rangle$	0.9983	-0.0577	0.0049	0.7325	0.9999	0.0120	0.0049	0.7316
$ y\rangle$	-1.0000	0.0072	0.0004	0.8929	-1.0000	0.0072	0.0004	0.8931
$ r\rangle$	0.0097	-0.0507	0.9987	0.7372	0.0103	-0.0155	0.9998	0.7363
$ l\rangle$	0.0060	-0.1249	-0.9921	0.7809	0.0091	-0.0901	-0.9959	0.7790

Table R1. Calculated polarization purity and polarization conversion efficiency of the four typical polarization states (i.e., $|x\rangle$, $|y\rangle$, $|r\rangle$, $|l\rangle$), with DWP oriented at respective $\theta_{\text{DWP}} = 45^\circ$ and 46° , for linearly $|y\rangle$ polarized light incidence.

3. As a final remark, I'm a bit doubtful about the comparison made in the supplementary table 1. The source of my doubt is that I am unsure how the authors have derived the efficiency in each of these experiments and if the comparison is fair. To evaluate the efficiency obtained in this manuscript the authors are using the degree of polarisation. Doing a quick search through the other cited works it seems to me that the numbers they provide in the table do not correspond to the degree of polarisation for all the other references and I think this might lead to a slightly unfair comparison. So I'd appreciate if the authors could clarify what they define as efficiency and use the same metric for the comparison of each experiment.

Our Response:

We thank the reviewer for this attentive comment. In this manuscript, the polarization conversion efficiency we used is the ratio of the reflected light power in specific polarization channels (i.e., $|x\rangle$, $|y\rangle$, $|r\rangle$, $|l\rangle$) to the incident $|y\rangle$ polarized light power.

For clarity, we added this definition in the main text (p. 11, revised manuscript):

“and the degree of polarization (DOP) is defined as $DOP = \sqrt{s_1^2 + s_2^2 + s_3^2}$. The polarization conversion efficiency (Fig. 2 and Supplementary Figs. S9–S11) is defined as the ratio of the reflected light power in a specific polarization channel (i.e., $|x\rangle$, $|y\rangle$, $|a\rangle$, $|b\rangle$, $|r\rangle$, $|l\rangle$) to the incident linearly $|y\rangle$ polarized light power.”

To make a fair comparison between our work and others, the efficiencies from Ref 1–3 summarized in the Supplementary Table 1 are estimated in a same way, and of course, with working point around the maximum birefringence-tunability region. However, this definition becomes problematic to use for the demonstrations exhibiting very small birefringence tuning ranges, for example, Ref. 4–6. In these cases, the complete polarization conversion between typical polarization states (i.e., $|x\rangle$, $|y\rangle$, $|a\rangle$, $|b\rangle$, $|r\rangle$, $|l\rangle$) is simply not possible, thus rendering the polarization conversion efficiencies being very small. For these experiments, we estimated the normalized reflectance or transmittance changes in one specific polarization channel instead to show their polarization conversion capabilities. To clarify this difference in the way the efficiency has been determined in the Supplementary Table 1, we added the corresponding explanatory text: (p.2, revised supplementary information):

“* The efficiency listed for Ref. 4 and 6 is derived from the normalized reflectance or transmittance change in one specific polarization channel, to indicate their polarization conversion capabilities.”

Other modifications:

- All changes in the main text are noted in highlighted text.
- Figure numbers in the revised Supplementary Information are updated, and corresponding changes are made in the main text.

REVIEWERS' COMMENTS

Reviewer #2 (Remarks to the Author):

I thank the authors for addressing my comments and I agree with them that the revised manuscript is now improved and ready for publication in Nature Communications.

REVIEWERS' COMMENTS

Reviewer #2 (Remarks to the Author):

I thank the authors for addressing my comments and I agree with them that the revised manuscript is now improved and ready for publication in Nature Communications.

Response to Reviewers

Reviewer #2 (Remarks to the Author):

I thank the authors for addressing my comments and I agree with them that the revised manuscript is now improved and ready for publication in Nature Communications.

Our Response:

We thank the reviewer for his/her recommendation of our work.